# A large-scale resource for tissue-specific CRISPR mutagenesis in *Drosophila*

**Fillip Port\*, Claudia Strein, Mona Stricker, Benedikt Rauscher, Florian Heigwer, Jun Zhou, Celine Beyersdörffer, Jana Frei, Amy Hess, Katharina Kern, Laura Lange, Nora Langner, Roberta Malamud, Bojana Pavlović, Kristin Rädecke, Lukas Schmitt, Lukas Voos, Erica Valentini, Michael Boutros\***

German Cancer Research Center (DKFZ), Division Signaling and Functional Genomics and Heidelberg University, Heidelberg, Germany

**Abstract** Genetic screens are powerful tools for the functional annotation of genomes. In the context of multicellular organisms, interrogation of gene function is greatly facilitated by methods that allow spatial and temporal control of gene abrogation. Here, we describe a large-scale transgenic short guide (sg) RNA library for efficient CRISPR-based disruption of specific target genes in a constitutive or conditional manner. The library consists currently of more than 2600 plasmids and 1700 fly lines with a focus on targeting kinases, phosphatases and transcription factors, each expressing two sgRNAs under control of the Gal4/UAS system. We show that conditional CRISPR mutagenesis is robust across many target genes and can be efficiently employed in various somatic tissues, as well as the germline. In order to prevent artefacts commonly associated with excessive amounts of Cas9 protein, we have developed a series of novel UAS-Cas9 transgenes, which allow fine tuning of Cas9 expression to achieve high gene editing activity without detectable toxicity. Functional assays, as well as direct sequencing of genomic sgRNA target sites, indicates that the vast majority of transgenic sgRNA lines mediate efficient gene disruption. Furthermore, we conducted the so far largest fully transgenic CRISPR screen in any metazoan organism, which further supported the high efficiency and accuracy of our library and revealed many so far uncharacterized genes essential for development.

**\*For correspondence:**
f.port@dkfz.de (FP);
m.boutros@dkfz.de (MB)

**Competing interests:** The authors declare that no competing interests exist.

## Introduction

The functional annotation of the genome is a prerequisite to gain a deeper understanding of the molecular and cellular mechanisms that underpin development, homeostasis and disease of multicellular organisms. *Drosophila melanogaster* has provided many fundamental insights into metazoan biology, in particular in the form of systematic gene discovery through genetic screens. Forward genetic screens utilize random mutagenesis to introduce novel genetic variants, but are limited by the large number of individuals required to probe many or all genetic loci and difficulties in identifying causal variants. In contrast, reverse genetic approaches, such as RNA interference (RNAi), are gene-centric designed and allow to probe the function of a large number of genes (*Boutros and Ahringer, 2008*; *Heigwer et al., 2018*; *Horn et al., 2011*; *Mohr et al., 2014*). In addition, RNAi reagents can be genetically encoded and used to screen for gene function with spatial and temporal precision (*Dietzl et al., 2007*; *Kaya-Çopur and Schnorrer, 2016*; *Ni et al., 2009*). However, RNAi is often limited by incomplete penetrance due to residual gene expression and can suffer from off-target effects (*Echeverri et al., 2006*; *Ma et al., 2006*; *Perkins et al., 2015*).

While genetic screens have contributed enormously to our understanding of gene function, large parts of eukaryotic genomes remain not or only poorly characterized (*Brown et al., 2009*; *Dickinson et al., 2016*; *White et al., 2013*). For example, in *Drosophila* only 20% of genes have associated mutant alleles (*Kaufman, 2017*). Therefore, there exists an urgent need to develop

**eLife digest** Twenty years after the release of the sequence of the human genome, the role of many genes is still unknown. This is partly because some of these genes may only be active in specific types of cells or for short periods of time, which makes them difficult to study.

A powerful way to gather information about human genes is to examine their equivalents in 'model' animals such as fruit flies. Researchers can use genetic methods to create strains of insects where genes are deactivated; evaluating the impact of these manipulations on the animals helps to understand the roles of the defunct genes. However, the current methods struggle to easily delete target genes, especially only in certain cells, or at precise times.

Here, Port et al. genetically engineered flies that carry CRISPR-Cas9, a biological system that can be programmed to 'cut' and mutate precise genetic sequences. The insects were also manipulated in such a way that the CRISPR elements could be switched on at will, and their quantity finely tuned. This work resulted in a collection of more than 1,700 fruit fly strains in which specific genes could be deactivated on demand in precise cells. Further experiments confirmed that this CRISPR system could mutate target genes in different parts of the fly, including in the eyes, gut and wings.

Port et al. have made their collection of genetically engineered fruit flies publically available, so that other researchers can use the strains in their experiments. The CRISPR technology they refined and developed may also lay the foundation for similar collections in other model organisms.

innovative approaches to gain a more complete understanding of the functions encoded by the various elements of the genome.

Clustered Regularly Interspersed Short Palindromic Repeats (CRISPR) - CRISPR-associated (Cas) systems are adaptive prokaryotic immune systems that have been adopted for genome engineering applications (*Doudna and Charpentier, 2014*; *Wang et al., 2016*). Cas9 complexed with a single chimeric guide RNA (sgRNA) mediates site-specific DNA double strand breaks and subsequent DNA repair can result in small insertions and deletions (indels) at the break point. However, not all Cas9-mediated indel mutations abrogate gene function. To compensate for that, strategies have been developed to introduce simultaneously several mutations in the same gene. The efficiency of such multiplexing strategies has been demonstrated in flies, mice, fish and plants, and several sgRNAs are often required to generate bi-allelic loss-of function mutations in all cells (*Port and Bullock, 2016*; *Xie et al., 2015*; *Yin et al., 2015*). Furthermore, to gain a comprehensive understanding of the often multifaceted functions genetic elements have in multicellular organisms requires methods that enable spatial or temporal control of gene disruption. To restrict CRISPR mutagenesis to defined cells, tissues or developmental stages, specific regulatory regions are commonly employed to drive Cas9 expression. However, Cas9 expression vectors with tissue-specific enhancers often display 'leaky' Cas9 expression in other tissues and poor control of CRISPR mutagenesis has been observed in multiple systems, including flies, mice and patient derived xenografts (*Chen et al., 2017*; *Dow et al., 2015*; *Hulton et al., 2019*; *Port and Bullock, 2016*). It has recently been demonstrated that expressing both Cas9 and sgRNA from conditional regulatory elements can result in tightly controlled genome editing (*Port and Bullock, 2016*), but the robustness of such a strategy across many genomic target sites has so far not been explored.

Here, we describe a large-scale resource for spatially restricted mutagenesis in *Drosophila*. The system mediates robust mutagenesis across target genes, giving rise to a large fraction of cells containing gene knock-outs and displays tight spatial and temporal control. We developed a series of tunable Cas9 lines that allow gene editing with high efficiency and low toxicity independent of enhancer strength. These can be used with a growing library of sgRNA transgenes, which currently comprise over 1700 *Drosophila* strains, for systematic mutagenesis in any somatic tissue or the germline. Furthermore, we present the first large-scale transgenic CRISPR screen using this resource, which confirms its high efficiency and specificity and reveals multiple uncharacterized genes with essential, but unknown function.

## Results

### Robust tissue-specific CRISPR mutagenesis

We set out to develop a large-scale resource that would allow systematic CRISPR-mediated gene disruption with tight spatial and temporal control (*Figure 1A*). In *Drosophila*, tissue-specific expression of transgenes is most commonly performed via the binary Gal4/UAS system (*Brand and Perrimon, 1993*) and thousands of Gal4 lines with specific temporal and spatial expression patterns are publicly available. To harness this resource for tissue-specific CRISPR mutagenesis we aimed to utilize *UAS-Cas9* transgenes and combine them with the sgRNA expression vector *pCFD6*, which enables Gal4-dependent expression of sgRNA arrays. We have previously shown that conditional expression of both Cas9 and sgRNAs is necessary to achieve tight control of mutagenesis (*Figure 1B*; *Port and Bullock, 2016*). Since this previous proof-of principle study was restricted to testing *pCFD6* with two sgRNAs targeting the Wnt secretion factor *Evenness interrupted* (*Evi*, also known as *Wntless* or *Sprinter*; *Bänziger et al., 2006*; *Bartscherer et al., 2006*; *Port and Bullock, 2016*), we first tested whether this system is robust across target genes and tissues, a prerequisite to generate large-scale libraries of sgRNA strains targeting many or all *Drosophila* genes. To this end, we created various transgenic fly lines harbouring a *pCFD6* transgene encoding two sgRNAs targeting a single gene at two independent positions. These were crossed to flies containing a *UAS-cas9.P2* transgene and a tissue-specific Gal4 driver. We then analysed if mutations were efficiently induced, restricted to the appropriate cells and caused the expected phenotypes. We observed efficient and specific gene disruption in wing imaginal discs with *pCFD6* sgRNA transgenes targeting the *Drosophila* beta-Catenin homolog *armadillo* (*arm*, *Figure 1C*), as well as the transcription factor *senseless* (*sens*) or the transmembrane protein *smoothened* (*smo*) (*Figure 1—figure supplement 1A,B*). To test tissue-specific CRISPR mutagenesis in a different tissue context, we targeted *Notch* (*N*) in the *Drosophila* midgut, which is derived from the endoderm. We observed a strong increase in stem cell proliferation and an accumulation of cells with small nuclei, which matches the described phenotype of *N* mutant clones in the midgut (*Ohlstein and Spradling, 2006; Figure 1D* and *Figure 1—figure supplement 2*). Interestingly, we observed a qualitative difference between perturbation of *N* expression by RNAi, which only induces hyperplasia in female flies (*Figure 1—figure supplement 2*; *Hudry et al., 2016*; *Siudeja et al., 2015*), and *N* mutagenesis by CRISPR, which induces strong overgrowth in both male and female midguts (*Figure 1—figure supplement 2*). We also tested conditional mutagenesis of *neuralized* (*neur*) and *yellow* (*y*) along the dorsal midline and of *sepia* (*se*) in the developing eye and observed in each case the described null mutant phenotype in the expected domain (*Figure 1E,F*, *Figure 1—figure supplement 1C*).

Next, we tested whether pCFD6-sgRNA$^{2x}$ also mediates efficient mutagenesis in the germline, where some UAS vectors are silenced (*DeLuca and Spradling, 2018*; *Huang et al., 2018*). This is a particularly important application, as it allows to create stable and sequence-verified mutant fly lines, which can be backcrossed to remove potential off-target mutations. We crossed previously described *nos-Gal4VP16 UAS-Cas9.P1* flies (*Port et al., 2014*) to sgRNA strains targeting either *neur*, *N*, *cut* (*ct*), *decapentaplegic* (*dpp*) or *Ras85D*. Despite the fact that all five genes are essential for *Drosophila* development and act in multiple tissues, *nos-Gal4VP16 UAS-Cas9.P1 pCFD6-sgRNA$^{2x}$* flies were viable and morphologically normal, demonstrating tightly restricted mutagenesis. We then tested their offspring for CRISPR induced mutations at the sgRNA target sites. Crosses with *pCFD6-sgRNA$^{2x}$* targeting *neur*, *N*, *ct* and *Ras85D* passed on mutations to most or all analysed offspring (*Figure 1G*). Mutations were often found on both target sites, were frequently out-of-frame and included large deletions of 8 and 14 kb between the sgRNA target sites (*Figure 1G*). In contrast, *nos-Gal4VP16 UAS-Cas9.P1 pCFD6-dpp$^{2x}$* flies produced only few viable offspring of which only 1/11 carried a mutation, which was in-frame. Since *dpp* is known to be haploinsufficient (*St Johnston et al., 1990*), this is consistent with a high number of *dpp* loss-of function alleles being transmitted to the next generation.

Together, these experiments demonstrate that sgRNA expression from *pCFD6* mediates efficient and tightly restricted mutagenesis in various somatic cell types as well as the germline and establishes that tissue-specific CRISPR mutagenesis in *Drosophila* is robust across genes and tissues.

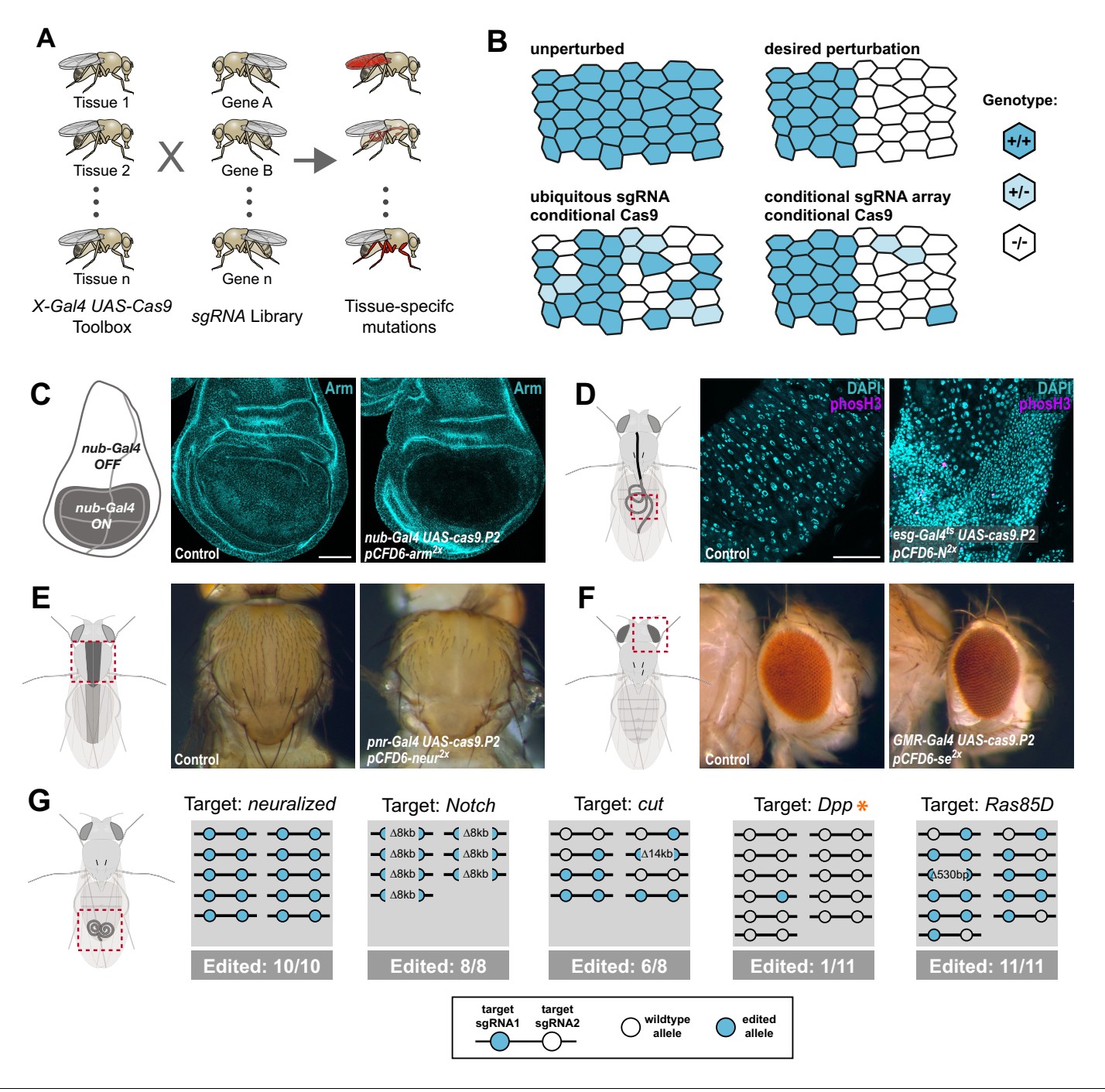

**Figure 1.** Conditional CRISPR mutagenesis with pCFD6 is robust across target genes and tissues. (**A**) Schematic overview of the workflow. To perform tissue-specific targeted mutagenesis flies transgenic for a specific Gal4 driver (*X-Gal4*) and *UAS-Cas9* are crossed to flies with a *UAS-sgRNA* transgene. Offspring from this cross express Cas9 and sgRNAs in Gal4 expressing cells, leading to mutagenesis of the target gene. (**B**) Schematic of gene editing outcomes typically observed with a single, ubiquitous sgRNA (lower left) or a conditional array of several sgRNAs (lower right). Leaky expression, that is expression in the absence of Gal4, from conditional Cas9 transgenes gives rise to ectopic mutagenesis in combination with ubiquitous, but not conditional, sgRNAs. Gene editing in tissues typically results in genetic mosaics, which can be enriched for bi-allelic knock-out cells through sgRNA multiplexing. (**C**) Conditional CRISPR mutagenesis in wing imaginal discs with *nub-Gal4* in the wing pouch. Gene editing with *pCFD6-arm*$^{2x}$ results in loss of Arm protein exclusively in the Gal4 expression domain in nearly all cells. Control animals express the *nub-Gal4* driver and *UAS-cas9.P2*. Scale bar = 50 μm. (**D**) Conditional CRISPR mutagenesis of *Notch* in intestinal stem cells drives tumor formation in the midgut. *esg*$^{ts}$ (*esg-Gal4 tub-Gal80*$^{ts}$) was used to repress expression of *UAS-cas9.P2* and *pCFD6-N*$^{2x}$ until adult stages. Mutagenesis was induced for 5 days at 29°C and flies were returned to 18°C to avoid Cas9.P2 mediated toxicity. Posterior midguts 15 days after induction of mutagenesis are shown. *esg*$^{ts}$ *UAS-cas9.P2 pCFD6-N*$^{2x}$ tissue

*Figure 1 continued on next page*

Figure 1 continued

shows an accumulation of stem cells (DNA marked in cyan) and an increase in mitotic cells (pHistone3 in magenta). Quantification of phenotypes are shown in **Figure 1—figure supplement 2**. Control genotype is *esg^ts UAS-cas9.P2 pCFD6-se^2x*. Scale bar = 50 µm. (**E**) Mutagenesis of *neur* in *pnr-Gal4 UAS-cas9.P2 pCFD6-neur^2x* animals results in loss of thoracic bristles along the dorsal midline, where *pnr-Gal4* is expressed. Note the tissue patch that retains bristles, reflecting mosaic mutagenesis. (**F**) Mutagenesis of the pigmentation gene *se* in the eye. *GMR-Gal4 UAS-casp.P2 pCFD6-se^2x* animals develop a uniform dark eye coloration. Control animals in (**E**) and (**F**) express the respective Gal4 driver and *UAS-cas9.P2 pCFD6-Sfp24C1^2x*. (**G**) *pCFD6* mediated mutagenesis in the germline. Shown is a summary of the mutational status at each sgRNA target site in individual F1 flies. *nos-Gal4VP16 UAS-cas9.P1 pCFD6* flies expressing sgRNAs targeting the indicated essential genes are viable, demonstrating germline restricted mutagenesis, and transmit mutant alleles to their offspring. Shown is a summary of the mutational status at each sgRNA target site in individual flies. All lines, except the one targeting *Dpp* (asterisk), transmit mutant alleles to the majority of offspring. Flies expressing sgRNAs targeting *Dpp* in the germline produce few viable offspring and transmitted only a single, in-frame, mutation out of 11 analysed offspring. The same sgRNA construct results in highly efficient mutagenesis in somatic tissues (see **Figure 4**), consistent with haploinsufficiency of *Dpp* in the *Drosophila* embryo.

The online version of this article includes the following figure supplement(s) for figure 1:

**Figure supplement 1.** Efficient conditional CRISPR mutagenesis in various *Drosophila* tissues.

**Figure supplement 2.** Qualitative differences between CRISPR mutagenesis and RNAi knock-down of *Notch* in the *Drosophila* midgut.

## Tunable Cas9 expression to balance activity and toxicity

We and others have shown that expression of high amounts of Cas9 protein is toxic in various organisms (**Jiang et al., 2014**; **Poe et al., 2019**; **Port et al., 2014**; **Yang et al., 2018**). For example, over-expression of Cas9 in the wing imaginal disc of *nub-Gal4 UAS-cas9.P2* animals results in a strong induction of apoptosis (**Figure 2—figure supplement 1A**). Since only relatively low levels of Cas9 are sufficient for efficient gene editing (**Figure 2—figure supplement 1B**), we sought to engineer a system that would allow to tune Cas9 expression to optimally balance activity and toxicity. Such a system would ideally allow to modulate Cas9 levels independent of enhancer strength, in order to be compatible with the wide range of available Gal4 lines. We employed a method that uses upstream open reading frames (uORF) of different length to predictably reduce translation of the main, downstream ORF (**Ferreira et al., 2013**; **Kozak, 2001**; **Southall et al., 2013**). We created a series of six *UAS-cas9* plasmids containing uORFs of different length, ranging from 33 bp (referred to as *UAS-u^XS Cas9*) to 714 bp (*UAS-u^XXL Cas9*, **Figure 2A**). When combined with *nos-cas9* these plasmids resulted in Cas9 protein levels inversely correlated with the length of the uORF (**Figure 2B**, **Figure 2—figure supplement 1C**). Reducing the amount of Cas9 protein resulted in a strong decrease in the number of apoptotic cells (**Figure 2C**). Importantly, three *UAS-uCas9* transgenes with moderate levels of Cas9 expression and apoptosis levels similar to controls did mediate full on-target gene editing activity at the *evi* locus in wing imaginal discs (**Figure 2D**, **Figure 2—figure supplement 1C**). Together, these experiments demonstrate that the UAS-uCas9 vector series enables titration of Cas9 expression to avoid toxicity without sacrificing gene editing activity.

Next, we generated a toolbox of various fly strains harbouring a *UAS-u^M Cas9* transgene and a Gal4 driver on the same chromosome (**Figure 2—figure supplement 2A,B**). Such stocks can be crossed to transgenic sgRNA lines to induce conditional CRISPR mutagenesis in Gal4-expressing cells. We tested the spatial mutagenesis pattern for a number of novel *Gal4 UAS-u^M Cas9* lines in the wing imaginal disc of third instar larva by either visualizing the loss of protein encoded by the target gene with a specific antibody, or by using the transgenic CIGAR reporter (**Brunner et al., 2019**). CIGAR encodes an ubiquitously expressed fluorescent protein that is only efficiently translated once an upstream sequence has been mutated by CRISPR gene editing (**Brunner et al., 2019**). While not all CRISPR-mediated mutations lead to induction of the fluorophore encoded by CIGAR, this strategy has the advantage that it readily reveals CRISPR activity throughout the entire organism.

We found that while some *Gal4 UAS-u^M Cas9* lines resulted in mutagenesis exclusively in cells positive for Cas9 at that stage (**Figure 2—figure supplement 2D,E**), others had much broader mutagenesis patterns (**Figure 2E**, **Figure 2—figure supplement 2F,G**). For example, in third instar wing discs *ptc-Gal4* is expressed in a narrow band of cells along the anterior-posterior boundary (**Figure 2E**). However, CRISPR mutagenesis with *ptc-Gal4* frequently leads to mutations throughout the entire anterior compartment (**Figure 2E'**), likely reflecting broader expression of *ptc-Gal4* in early development or expression at low level in this domain. Similar effects were observed with *dpp-Gal4* (**Figure 2—figure supplement 2G**). Therefore, additional regulatory mechanisms to temporally control Cas9 expression are highly desirable when using Gal4 lines with dynamic expression patterns

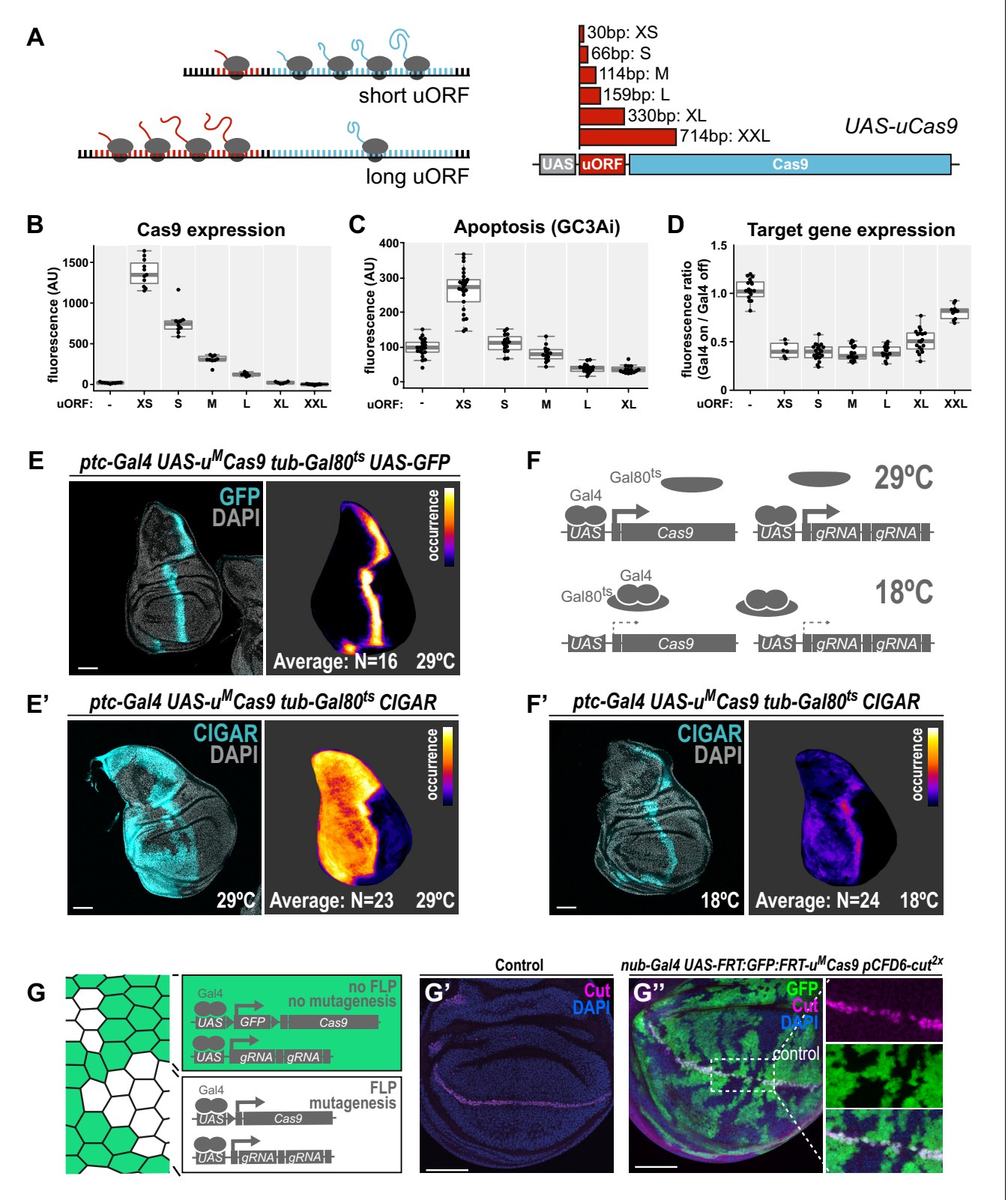

**Figure 2.** A transgenic series for tunable Cas9 expression to balance activity and toxicity. (A) Principle of the UAS-uCas9 series. Translation of the downstream ORF is inversely correlated with length of the upstream ORF in bicistronic mRNAs. The UAS-uCas9 series consists of transgenes that harbor uORFs of different length to modulate expression of Cas9. (B - D) Systematic characterization of Cas9 expression, toxicity and mutagenesis efficiency of the UAS-uCas9 series. Transgenes of the UAS-uCas9 series were recombined with nub-Gal4 and crossed to the apoptosis sensor UAS-

*Figure 2 continued on next page*

*Figure 2 continued*

GC3Ai (B, C) or pCFD6-evi2x (D). Graphs show data as individual dots, and boxplots as a data summary, with the line representing the median and the box the interquartile range. (B) Quantification of anti-Cas9 staining intensity in wing discs of the indicated genotype. Cas9 levels gradually reduce as the size of the uORF increases. N ≥ 6 wing discs. (C) Elevated levels of apoptosis were only observed with UAS-uXSCas9. The longest uORF (uXXL) encodes EGFP, preventing visualization of dying cells with GC3Ai. Quantification of fluorescent intensity of the GC3Ai reporter in the wing pouch. N ≥ 14 wing discs. (D) All transgenes of the UAS-uCas9 series mediate evi mutagenesis, with transgenes containing the four shortest uORFs (XS-L) leading to comparable gene editing that removes Evi from nearly all cells in the Gal4 expression domain. Quantification of staining intensity for Evi protein in the wing pouch (Gal4 on), relative to Evi staining in the hinge region (Gal4 off). N ≥ 6 wing discs. (E, E') CRISPR mutagenesis patterns reflect Gal4 expression history. (E) Fluorescence of GFP, which turns over, reflects most recent Gal4 expression pattern. (E') CRISPR mutagenesis, visualized by activation of the CIGAR reporter, is permanent and reveals the Gal4 expression history. Images of a representative wing disc are shown to the left of each panel and average intensity projection of several discs registered to a common template are shown on the right (see Materials and methods). Areas that are CIGAR positive in many discs appear bright, while areas devoid of signal in most discs appear dark. (F, F') Incomplete repression of CRISPR mutagenesis by temperature-sensitive Gal80. (F) Principle of the Gal80ts system. At 18°C Gal80 binds and inhibits Gal4. (F') Mutagenesis is still observed at 18°C in 11/24 discs and observed preferentially in the Gal4 expression domain, indicating incomplete Gal4 suppression by Gal80ts. (G - G'') Control of CRISPR mutagenesis by a flip-out cassette. (G) In the absence of FLP recombinase a FRT-flanked GFP flip-out cassette (FRT sites represented by triangles) separates Cas9 from the promoter, resulting in cells that express GFP, but no Cas9. In the presence of FLP, the GFP cassette is excised and Cas9 is expressed. (G') Staining for the transcription factor Cut reveals a continuous stripe of cells expressing ct along the dorsal-ventral boundary in wildtype tissue. (G'') A pulse of FLP expression was used to excise the GFP flip-out cassette in a subset of cells (marked by the absence of GFP). Cut expression (magenta) is exclusively lost in GFP negative cells. Scale bar = 50 μm.

The online version of this article includes the following figure supplement(s) for figure 2:

**Figure supplement 1.** High levels of Cas9 expression from *UAS-cas9*.
**Figure supplement 2.** CRISPR mutagenesis patterns reflect expression patterns of Gal4 lines throughout development.

during development. We first employed the temperature-sensitive Gal80 repressor to suppress Gal4 activity. While Gal80^ts mediated strong inhibition of mutagenesis in *ptc-Gal4 UAS-uMCas9 tub-Gal80^ts* flies at the restrictive temperature of 18°C, we still observed mutagenesis in Gal4-expressing cells in 11/24 wing discs, indicating residual Gal4 activity (*Figure 2F*). We therefore tested an alternative strategy to induce CRISPR mutagenesis at a given time point. We created a transgene that harbors a FRT-flanked GFP Stop-cassette between the *UAS* promoter and the *u^MCas9* expression cassette (*UAS-FRT-GFP-FRT-u^MCas9*, *Figure 2G*). A brief pulse of Flp recombinase (from a *hs-Flp* transgene) can be used to excise the GFP cassette at the desired time and induce Cas9 expression. We validated this approach by mutagenizing *ct* in a negatively marked subset of cells in the wing disc and observed loss of Ct protein exclusively in cells that had lost GFP expression (*Figure 2G*).

These experiments highlight the need to critically evaluate spatial mutagenesis patterns in conditional CRISPR experiments and suggest strategies for additional control of gene editing in cases where the Gal4 expression pattern alone does not suffice. We envision that in the future other systems for conditional transgene expression, such as the chemical-dependent GeneSwitch system (*Osterwalder et al., 2001*; *Roman et al., 2001*), split-Gal4 (*Luan et al., 2006*) or conditional transgene degradation (*Sethi and Wang, 2017*) will also be combined with CRISPR to further refine mutagenesis patterns.

## A large-scale transgenic sgRNA library

Having established the robustness of our method and developed an optimised Cas9 toolkit, we next focused our efforts on the generation of a large-scale sgRNA resource. First, we generated and validated three sgRNA lines targeting genes with highly restricted expression patterns, which can be used as controls for effects of Cas9/sgRNA expression and induction of DNA damage in the majority of tissues where their target gene is not expressed (*Figure 3—figure supplement 1*; *Graveley et al., 2011*). To allow systematic screening of functional gene groups we then designed sgRNAs against all *Drosophila* genes encoding transcription factors, kinases and phosphatases, as well as a large number of other genes encoding fly orthologs of genes implicated in human pathologies (*Figure 3A*, see Materials and methods). We used CRISPR library designer (*Heigwer et al., 2016*) to compile a list of all sgRNAs that do not have predicted off-target sites elsewhere in the genome. We then selected sgRNAs depending on the position of their target site within the target gene. We chose sgRNAs targeting coding exons shared by all mRNA isoforms and target sites that were located in the 5' half of the open reading frame, where indel mutations often have the largest functional impact. We then grouped sgRNAs in pairs, with each pair targeting sites typically

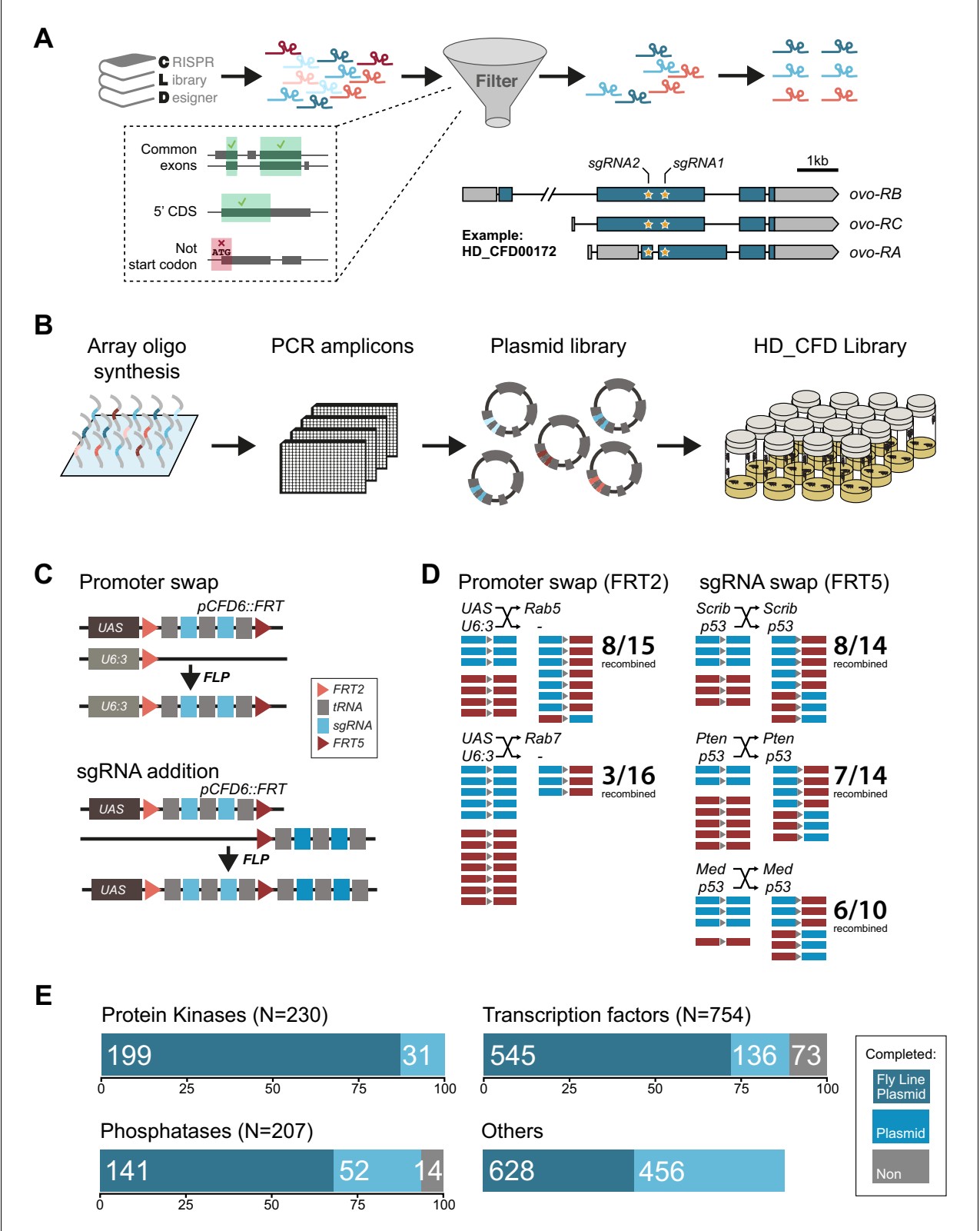

**Figure 3.** Generation of a large-scale sgRNA library. (**A**) Design of the sgRNA pairs used for the HD_CFD library. sgRNAs were designed through CLD and filtered to target common exons in the 5'ORF and not overlap the start codon. sgRNAs were then paired to target two independent positions in the same gene. As an example the locations of the two target sites in *ovo* targeted by the two sgRNAs encoded in line HD_CFD000172 is shown. Exons are represented as boxes and regions in blue are protein coding. (**B**) Experimental strategy for the generation of the transgenic sgRNA library. *Figure 3 continued on next page*

*Figure 3 continued*

sgRNA target sequences are encoded on oligonucleotides synthesized and cloned in pool. Individual plasmids are sequence verified and transformed into *Drosophila* at attP40 on the second chromosome following a pooled injection protocol followed by genotyping of individual transformants. (C) Applications of the *pCFD6::FRT* vector. *pCFD6::FRT* contains two non-compatible FRT sites either side of the sgRNA cassette. Using compatible FRT sides in trans allows to exchange sequences upstream or downstream of the sgRNAs *in vivo*. (D) Efficient promoter or sgRNA exchange *in vivo*. Summary of FLP/FRT mediated exchange of the sgRNA promoter (left) or sgRNAs (right). Each line represents a single sequenced animal. Red and blue boxes either side of the triangle (representing FRT) indicate successful recombination. (E) Summary statistics of the different functional groups present in the sgRNA library. Given is the number of genes from each category that are covered by fly lines, plasmids or against which currently no tools are available. Note that for some genes two fly lines or plasmids exist. Status in September 2019 is shown. Group 'Others' contains mainly genes with human orthologs associated with cancer development in humans.

The online version of this article includes the following figure supplement(s) for figure 3:

**Figure supplement 1.** Negative control sgRNA transgenes for use with HD_CFD library.

---

separated by approximately 500 bp of protein coding DNA (see Materials and methods). Next, we devised an efficient cloning protocol to insert defined sgRNA pairs into *pCFD6*. This utilized synthesized oligonucleotide pools, which allow cloning of hundreds to thousands of sgRNA plasmids in parallel in a single tube, followed by clonal selection of individual pCFD6-sgRNA$^{2x}$ plasmids and sequence validation (*Figure 3B*, see Materials and methods). We also generated a derivative of *pCFD6*, *pCFD6.FRT*, which harbors incompatible FRT2 and FRT5 sites before and after the sgRNA cassette, respectively. These recombination sites can be used to exchange sequences either side of the sgRNA cassette, for example the promoter, or to add additional sgRNAs to the array (*Figure 3C*). We validated that both FRT sites mediate highly efficient chromosome exchange *in vivo* (*Figure 3D*). We then generated a large-scale transgenic sgRNA library, which we collectively refer to as the 'Heidelberg CRISPR Fly Design Library' (short HD_CFD library). This growing resource currently contains 2622 plasmids and 1739 fly stocks targeting 1513 unique genes (*Supplementary file 1*). Fly lines are so far available for 545/754 (72%) transcription factors, 199/230 (87%) protein kinases and 141/207 (68%) phosphatases (*Figure 2D*).

## HD_CFD sgRNA lines mediate efficient mutagenesis and allow robust CRISPR screening

To test on-target activity of HD_CFD sgRNA strains, we crossed a random selection of 28 HD_CFD lines to an *act-cas9;;tub-Gal4/TM3* strain, which is expected to mediate ubiquitous mutagenesis in combination with active sgRNAs. We then sequenced PCR amplicons encompassing the sgRNA target sites (see Materials and methods) and analysed editing efficiency by ICE analysis (*Hsiau et al., 2019*). We found that the vast majority (26/28) of HD_CFD sgRNA lines resulted in gene editing on both target sites (*Figure 4A*). For 12/28 of lines editing on both sites was inferred to be at least 50% and 23/28 reached this threshold on at least one target site. In contrast, only a single line (HD_CFD00032) resulted in no detectable gene editing at either sgRNA target site. This suggests that HD_CFD sgRNA lines mediate robust and efficient mutagenesis of target genes across the genome.

Next, we performed a large-scale transgenic CRISPR screen. We crossed HD_CFD animals to *act-cas9;;tub-Gal4/TM3* to induce mutations ubiquitously in the offspring and determined viability at five to seven days after eclosion. 290/639 (45%) of all crosses did not yield any viable offspring, while 269 (42%) lines produced viable adults and 53 (8%) of the lines resulted in lethality with incomplete penetrance (*Figure 4B* and *Supplementary file 2*). In order to benchmark the performance of the screen, we manually curated viability information based on genetic alleles stored in the Flybase database to determine which HD_CFD lines target genes known to be essential or non-essential during *Drosophila* development. This resulted in a list of 210 lines that target known essential genes. Of those, 167 (79%) resulted in lethality, 20 (10%) were scored as semi-lethal, and 23 (11%) gave rise to viable adult offspring. Interestingly, among the targets of sgRNA lines that produced false-negative results there was a strong enrichment of genes known to play important roles, and to be highly expressed, during early embryonic development. Furthermore, sequencing the sgRNA target sites in randomly selected false-negative lines revealed efficient gene editing on one or both sites in 3/3 lines (*Figure 4—figure supplement 1*), suggesting that false-negative results often arise due to mRNA perdurance, not inactive sgRNAs. Next, we analysed our data set for the occurrence of false-

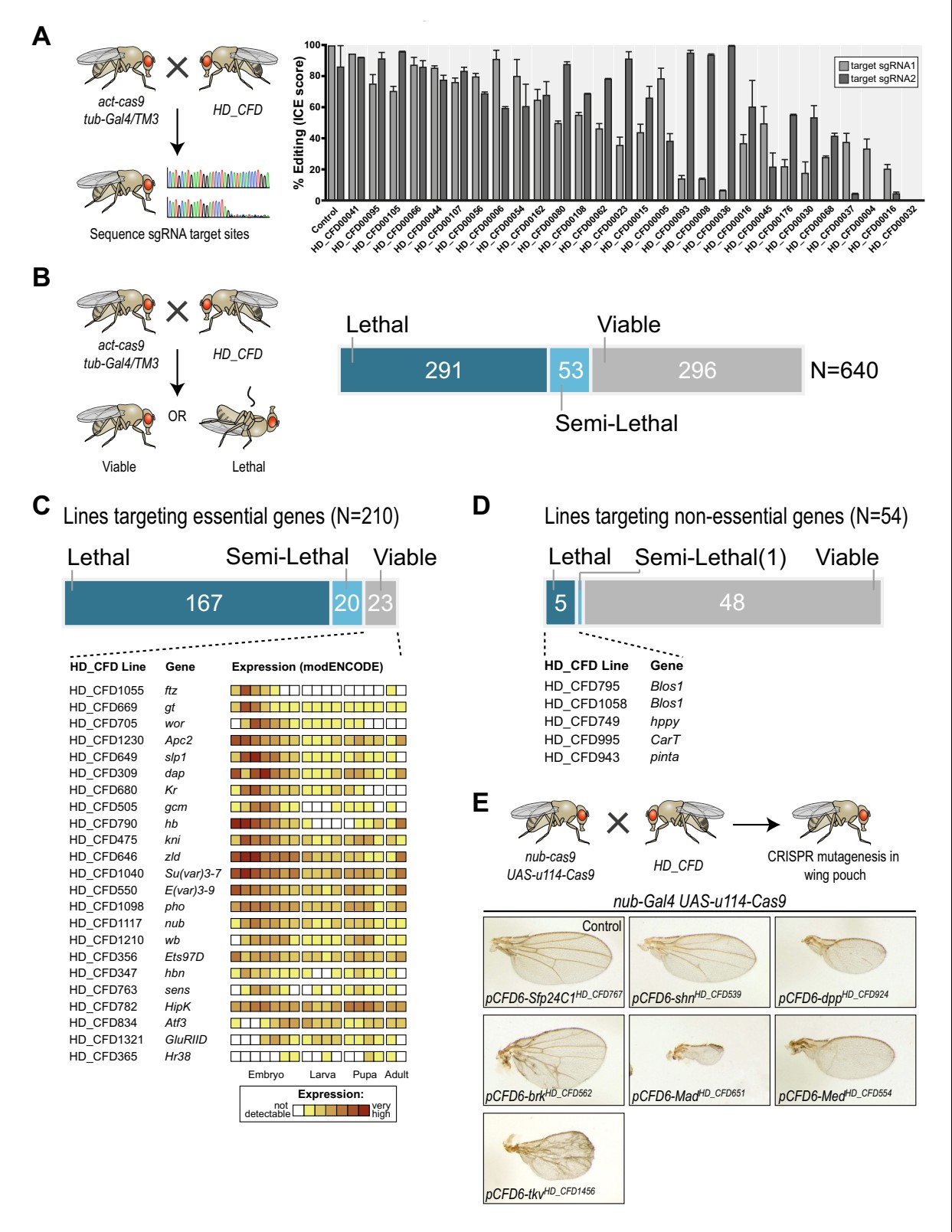

**Figure 4.** A large-scale CRISPR screen for essential genes in *Drosophila*. (**A**) The majority of sgRNA lines mediates efficient mutagenesis on both sgRNA target sites. sgRNA transgenes were combined with *act-cas9* and *tub-Gal4* to induce ubiquitous mutagenesis. Mutagenesis was measured by sequencing PCR amplicons spanning the target sites followed by Inference of CRISPR Edits (ICE) analysis. Shown are mean values of 2–4 independent experiments and the standard error of the mean. (**B**) CRISPR screening for essential genes in *Drosophila*. Ubiquitous mutagenesis was induced in

*Figure 4 continued on next page*

*Figure 4 continued*

offspring of HD_CFD sgRNA lines crossed with *act-cas9;;tub-Gal4/TM6B* partners. Vials were analysed after 15–17 days (~5–7 days after eclosion) for viable *act-cas9;pCFD6-sgRNA^2x; tub-Gal4* offspring. Summary statistics are shown on the right. Crosses were scored as semi-lethal when flies of the correct genotype were present, but <50% of the number of TM6B flies, and dead larva, pupae or adults were evident in the vail. (**C**) False-negative results are rare and often occur for genes controlling early development. Summary statistics for 208 HD_CFD sgRNA lines targeting known essential genes are shown. 23 (11%) lines give rise to the incorrect (viable) phenotype. mRNA expression data for these target genes is shown below (data from modENCODE). Most genes have maternally contributed mRNA, are highly expressed in early embryonic stage or play known roles in embryonic development. (**D**) Low number of false-positive results caused by HD_CFD sgRNA lines. 54 HD_CFD lines in the screen target genes known to be dispensable for fly development. five lines result in lethality when crossed to *act-cas9;;tub-Gal4/TM6B* flies. Note that lines HD_CFD795 and HD_CFD1058 target the same gene with independent sgRNAs. (**E**) Tissue-specific CRISPR mutagenesis in the developing wing. Representative images of adult wing phenotypes caused by CRISPR mutagenesis of Dpp signalling components are shown. All lines give rise to the expected alterations in wing size and vein patterning with varying strength.

The online version of this article includes the following figure supplement(s) for figure 4:

**Figure supplement 1.** HD_CFD sgRNA line resulting in false-negative results mediate efficient on-target mutagenesis.

positives, that is lines that target non-essential genes, but result in lethality. Among the 639 lines present in our screen, 54 target genes are currently annotated as viable. Of those, 48 (89%) gave rise to viable adult offspring, one resulted in semi-lethal offspring and 5 (9%) produced no viable offspring. False-positive results might arise due to off-target mutagenesis, mutations that affect neighbouring genes or cis-elements located at the target-locus, or reflect incorrect annotations in the database. Of the five lines giving rise to false-positive results in our screen two target the same gene (*Blos1*), arguing against sgRNA-mediated off-target mutagenesis in this case.

Screening for lethality not only allowed us to benchmark our sgRNA library, but also revealed multiple lines targeting uncharacterized genes with putative essential functions (*Supplementary file 3*). For example, sgRNA line HD_CFD558 targets CG9890, an evolutionary conserved (55% amino acid similarity to the human ortholog) zinc finger protein of unknown function. Another interesting example is CG6470, which is targeted by HD_CFD557 and HD_CFD599 with independent sgRNAs. CG6470 encodes an uncharacterized zinc finger protein that despite its essential role during development is evolutionary restricted to the genus *Drosophila*. These examples highlight the value of our lethality screen beyond benchmarking our technology. To further characterize genes of interest sgRNA lines can then be used for tissue-specific mutagenesis, where genes performing similar cellular functions often give rise to phenotypes with high similarity. To demonstrate this application, we crossed several lines targeting genes associated with *dpp*/TGFb signalling with *nub-Gal4 UAS-u^MCas9* flies, which drive CRISPR mutagenesis in selected tissues, including cells giving rise to the adult wing. All these lines result in lethality in combination with a ubiquitous CRISPR system (*Supplementary file 2*), but gave rise to viable adults in combination with *nub-Gal4 UAS-u^MCas9*, highlighting again the tight control of mutagenesis. Moreover, all lines resulted in offspring that had wings of abnormal size and morphology and faithfully recapitulated the known phenotypes of loss-of function mutations of their target genes (*Figure 4E*). Together these results show that lines of the HD_CFD library can be used for systematic CRISPR screens *in vivo* and mediate relevant phenotypes with very high penetrance and specificity.

## Discussion

Here, we present a large-scale collection of transgenic sgRNA strains for conditional CRISPR mutagenesis in *Drosophila*. In combination with the associated toolbox of novel Cas9 constructs, the sgRNA lines mediate efficient mutagenesis with precise temporal and spatial control. This allows the rapid targeted disruption of genes in various contexts in the intact organism. The high performance of this resource relies on a) use of conditional sgRNA constructs to achieve strong dependency of CRISPR mutagenesis on Gal4, b) tunable Cas9 expression to achieve high on-target activity with low toxicity, c) the use of two sgRNAs targeting independent positions in the same gene to increase the fraction of cells that harbor non-functional mutations in both alleles. We validate our library by conducting a fully transgenic CRISPR mutagenesis screen, to our knowledge the so far largest in any multicellular animal, which revealed 259 putative essential genes, of which 56 are poorly characterized.

To date RNAi is the most commonly used method to disrupt gene expression in defined cell types or developmental stages in vivo. In *Drosophila*, transgenic RNAi libraries that cover most protein coding genes have been described (*Dietzl et al., 2007*; *Heigwer et al., 2018*; *Perkins et al., 2015*). However, a significant number of these lines do not mediate efficient gene knock-down and the majority reduces mRNA levels by less than 75% (*Perkins et al., 2015*). Residual gene expression can therefore mask phenotypes in RNAi experiments, which loss-of function alleles induced by CRISPR mutagenesis might reveal. In support of this notion three recent studies demonstrate that CRISPR mutagenesis *in vivo* can cause phenotypes that are significantly more penetrant than RNAi (*Meltzer et al., 2019*) or are missed altogether in RNAi experiments (*Delventhal et al., 2019*; *Schlichting et al., 2019*). Furthermore, our molecular analysis of mutations induced by the CRISPR library described here, as well as the phenotypes arising from them, suggest that the fraction of lines that produce no or only insufficient on-target mutations is less than 10%, which compares favorably to current *Drosophila* RNAi libraries. Together these observations strongly suggest that screening biological processes of interest by conditional CRISPR mutagenesis can reveal novel gene functions that have so far been missed in RNAi based experiments.

For conditional CRISPR mutagenesis to be broadly applicable, it needs to be effective in a wide range of tissues and cell types. We show here that our system works effectively in a number of important tissues in the fly, such as imaginal discs, the adult midgut and the germline. Furthermore, others have shown that CRISPR with transgenic components is also effective in postmitotic neurons (*Delventhal et al., 2019*) or in cells of the prothoracic gland, which are endoreplicating and contain multiple copies of the genome (*Huynh et al., 2018*). In addition, our finding that sequencing of PCR amplicons generated from genomic DNA of entire flies frequently indicates mutations in over 90% of the amplicons suggests that CRISPR is effective in most cells of the animal.

In parallel to the CRISPR library described here, the National Institute of Genetics (NIG) in Japan, the Transgenic RNAi Project (TRIP) at Harvard University and the Schuldiner group at the Weizmann Institute are generating collections of transgenic sgRNA lines (*Meltzer et al., 2019*; *Zirin et al., 2020*; https://shigen.nig.ac.jp/fly/nigfly/). These projects follow different strategies to prioritise target genes and hence the overlap between different collections is currently limited. Furthermore, there exist significant differences in design between these resources and the library described here. First, the NIG and the majority of TRIP and Weizmann libraries encode a single sgRNAs per transgene, while all HD_CFD lines encode two sgRNAs. Co-expression of more than one sgRNA against the same target leads to more penetrant phenotypes and reduces the number of inactive lines (*Port and Bullock, 2016*; *Xie et al., 2015*; *Yin et al., 2015*). Second, the HD_CFD sgRNAs are encoded in *pCFD6* or *pCFD6.FRT*, which are conditional UAS vectors, while all other libraries so far used plasmids expressing sgRNAs from ubiquitous U6 promoters. We have previously shown that expression of U6-sgRNA in combination with UAS-Cas9 alone is not sufficient to efficiently restrict mutagenesis to Gal4 expressing cells and that expression of sgRNAs from a UAS vector, such as *pCFD6*, results in a significant improvement in spatial and temporal control (*Port and Bullock, 2016*). The use of transgenes of the UAS-uCas9 series can reduce, but not prevent, unwanted mutagenesis in combination with U6-sgRNAs, as leaky (i.e. Gal4 independent) Cas9 expression is reduced in the presence of a uORF. An advantage of U6-sgRNA vectors is the consistent high sgRNA expression, whereas the level of sgRNAs expressed from UAS promoters depends on the strength of the Gal4 line and can become limiting with weak Gal4 drivers (*Meltzer et al., 2019*). Of note, *pCFD6. FRT* can alleviate this problem, as users can easily swap the UAS promoter for a U6:3 promoter in cases where high sgRNA expression is a higher priority than tight conditional mutagenesis. The different sgRNA libraries that are currently being developed are therefore complementary resources for CRISPR mutagenesis. Large-scale screens in different contexts using lines from different libraries will be informative about the optimal use of each resource.

Two decades after the publication of the genome sequence of humans, mice, flies, worms and many other organisms, the functional annotation of these genomes are still far from complete. CRISPR-Cas genome editing is accelerating the rate at which new gene functions are described. The resources described here will facilitate context-dependent functional genomics in *Drosophila*. New insights into the function of the fly genome will inform the functional annotation of the human genome, reveal conserved principles of metazoan biology and suggest control strategies for insect disease vectors.

## Materials and methods

### Plasmid construction

PCRs were performed with the Q5 Hot-start 2x master mix (New England Biolabs (NEB)) and cloning was performed using the In-Fusion HD cloning kit (Takara Bio) or restriction/ligation dependent cloning. Newly introduced sequences were verified by Sanger sequencing. Oligonucleotide sequences are listed in *Supplementary file 4*.

### UAS-uCas9 plasmids

The UAS-uCas9 series of plasmids was generated using the pUASg.attB plasmid backbone (*Bischof et al., 2013*). The plasmid was linearized with EcoRI and XhoI and sequences coding for mEGFP(A206K) and hCas9-SV403'UTR were introduced by In-Fusion cloning using standard procedures. Coding sequences for mEGFP(A206K) were ordered as a gBlock from Integrated DNA Technologies (IDT) and amplified with primers mEGFPfwd and mEGFPrev (*Supplementary file 4*). The sequence coding for *SpCas9* and an SV40 3'UTR were PCR amplified from plasmid *pAct-Cas9* (*Port et al., 2014*) with primers Cas9SV40fwd and Cas9SV40rev. Both PCR amplicons and the linearized plasmid backbone were assembled in a single reaction to generate plasmid *UAS-u$^{XXL}$Cas9*. UAS-uCas9 plasmids with shorter uORFs were generated by PCR amplification using *UAS-u$^{XXL}$Cas9* as template and the common fwd primer uCas9fwd in combination with rev primers binding at various positions in the mEGFP ORF (uXSCas9rev for *UAS-u$^{XS}$Cas9*; uSCas9rev for *UAS-u$^{S}$Cas9*; uMCas9rev for *UAS-u$^{M}$Cas9*; uLCas9rev for *UAS-u$^{L}$Cas9*; uXLCas9rev for *UAS-u$^{XL}$Cas9*). PCR products were cirularized by In-Fusion cloning and the sequence between the hsp70 promoter and the attP site was verified by Sanger sequencing. The UAS-uCas9 plasmid series and the full sequence of each plasmid will become available from Addgene (Addgene plasmids 127382–127387).

### UAS-FRT-GFP-FRT-u$^{M}$Cas9

To generate *UAS-FRT-GFP-FRT-u$^{M}$Cas9* plasmid *UAS-Cas9.P2* (*Port and Bullock, 2016*) was digested with EcoRI and the plasmid backbone was gel purified. The FRT-GFP-FRT cassette was ordered as two separate gBlocks from IDT (GFPflipout5 and GFPflipout3) and individually PCR amplified with primers GFPflipout5fwd and GFPflipout5rev or GFPflipout3fwd and GFPflipout3rev and gel purified. The two amplicons were mixed at equalmolar ratios and fused by extension PCR, adding primers GFPflipout5fwd and GFPflipout3rev after 8 PCR cycles for an additional 25 cycles. The final FRT-GFP-FRT cassette was gel purified. The u$^{M}$Cas9EcoRI fragment was PCR amplified from plasmid *UAS-u$^{M}$Cas9* with primers uMCas9EcoRIfwd and uMCas9EcoRIrev and gel purified. The plasmid backbone, FRT-GFP-FRT cassette and u$^{M}$Cas9EcoRI fragment were assembled by In-Fusion cloning and sequence from the first FRT site to the end of Cas9 was verified by Sanger sequencing. The *UAS-FRT-GFP-FRT-u$^{M}$Cas9* plasmid and the full sequence will become available from Addgene (Addgene plasmid 127388).

### pCFD6.FRT

*pCFD6.FRT* was generated as a derivative of *pCFD6*. *pCFD6* was linearized by restriction digestion with EcoRI-HF and XbaI. The sgRNA cassette was exchanged with a new cassette encoding (from 5' to 3'): 5'UTR spacer, FRT2 site, *D. mel.* tRNA Gly, BbsI site, sgRNA core, *D. mel.* tRNA Glu, BbsI site, sgRNA core, *Os. sat.* tRNA, FRT5 site. The new sgRNA cassette was ordered as a gBlock from IDT and cloned into the linearized pCFD6 plasmid and newly introduced sequences were verified by Sanger sequencing. pCFD6.FRT will become available from Addgene.

### sgRNA design

All possible sgRNA sequences targeting all transcription factors, kinases, phosphatases and a number of other - mostly disease relevant - genes in the *D. melanogaster* genome version BDGP6 were identified using the CRISPR library designer (CLD) software version 1.1.2 (*Heigwer et al., 2016*). CLD excludes sgRNA sequences that have predicted off-target sites elsewhere in the genome. The resulting pool of sequences was further filtered according to additional criteria. Specifically, sequences with BbsI and BsaI restriction sites were excluded. In addition, sequences containing stretches of 4 or more identical nucleotides were removed from the pool. Two pairs of sgRNAs targeting each

gene were then selected using a random sampling approach. For each gene, up to 10,000 pairs of sgRNA sequences were selected at random from the pool of available sequences. Each sequence pair was then evaluated according to a custom scoring function. In order to preferentially select sgRNA pairs that target constitutive exons, the scoring function awarded bonus points for each transcript targeted by either of the sgRNAs. Bonus points were further given to sgRNAs targeting the first half of the gene and small distances to the gene's transcription start site were awarded additionally. To avoid selecting pairs of overlapping sgRNAs that could potentially interfere with each other's activity, sgRNA pairs that were less than 75 bp apart from each other were strongly penalized. Further, sgRNAs targeting the gene within 500 bp of each other were penalized. This was done to avoid functional protein products in cases where the second sgRNA might correct an out-of-frame mutation introduced by the first sgRNA. Finally, we penalized sgRNA with predicted off-target effects according to CLD. The two top-scoring pairs for each gene were selected for the HD_CFD library.

## sgRNA library cloning

sgRNA pairs were cloned into BbsI digested pCFD6 (*Port and Bullock, 2016*) following a two-step pooled cloning protocol. Oligonucleotide pools were ordered from Twist Biosciences and Agilent Technologies. Each oligonucleotide contained two sgRNA protospacer sequences targeting the same gene separated by a BsaI restriction cassette. Furthermore, oligos contained sequences at either end for PCR amplification and BbsI sites at the 5' end of the first and 3' end of the second protospacer. An annotated example oligo is shown in *Supplementary file 4*. Oligo pools were resuspended in sterile dH$_2$O and amplified by PCR with primers Libampfwd and Libamprev, followed by BbsI digestion and gel purification. Digested oligo pools were then ligated into BbsI digested pCFD6 plasmid backbone, transformed into chemically competent bacteria and plated on agarose plates containing Carbenicillin. After incubation overnight at 37°C transformed bacteria were resuspended and plasmid DNA was extracted and digested with BsaI. Next, the sgRNA core sequence and tRNA required between the two protospacers, but not encoded on the oligos, were introduced. These were PCR amplified from pCFD6 using primers Core_tRNAfwd and Core_tRNArev. PCR amplicons were digested with BsaI and ligated into the BsaI digested pCFD6 plasmid pool containing the library oligos, transformed into chemically competent bacteria and plated on agarose plates containing Carbenicillin. The next day single colonies were picked and used to inoculate liquid cultures. The following day plasmid DNA was extracted and the sgRNA cassette was sequenced with primer pCFD6seqfwd2 to determine which oligo was inserted and to verify the sequence. Individual sequence verified pCFD6-sgRNA2x plasmids were stored at −20°C and make up the HD_CFD plasmid library.

## *Drosophila* strains and culture

Transgenic *Drosophila* strains used or generated in this study are listed in *Supplementary file 5*. Unless specified otherwise flies were kept at 25°C with 50 ± 10% humidity with a 12 hr light/12 hr dark cycle.

## Transgenesis

Transgenesis was performed with the PhiC31/attP/attB system and plasmids were inserted at landing site (P{y[+t7.7]CaryP}attP40) on the second chromosome. Additional insertions of *UAS-u$^M$Cas9* were generated at (M{3xP3-RFP.attP}ZH-51D) on the second chromosome and (M{3xP3-RFP.attP} ZH-86Fb) and (PBac{y+-attP-3B}VK00033) on the third chromosome. Microinjection of plasmids into *Drosophila* embryos was carried out using standard procedures either in house, or by the *Drosophila* Facility, Centre for Cellular and Molecular Platforms, Bangalore, India (http://www.ccamp.res.in/dro-sophila) or by the Fly Facility, Department of Genetics, University of Cambridge, UK (www.flyfacility. gen.cam.ac.uk/). Transgenesis of sgRNA plasmids was typically performed by a pooled injection protocol, as previously described (*Bischof et al., 2013*). Briefly, individual plasmids were pooled at equimolar ratio and DNA concentration was adjusted to 250 ng/μl in dH$_2$O. Plasmid pools were microinjected into y[1] M{vas-int.Dm}ZH-2A w[*]; (P{y[+t7.7]CaryP}attP40) embryos, raised to adulthood and individual flies crossed to P{ry[+t7.2]=hsFLP}1, y[1] w[1118]; Sp/CyO-GFP. Transgenic offspring was identified by orange eye color and individual flies crossed to P{ry[+t7.2]=hsFLP}1, y[1] w

[1118]; Sp/CyO-GFP balancer flies. In the very rare case that a plasmid stably inserted at a genomic locus different than the intended attP40 landing site, this typically resulted in a noticeably different eye colouration and such flies were discarded.

## Genotyping of sgRNA flies

Transgenic flies from pooled plasmid injections were genotyped to determine which plasmid was stably integrated into their genome. If transgenic flies were male or virgin female, animals were removed from the vials once offspring was apparent and prepared for genotyping. In the case of mated transgenic females, genotyping was performed in the next generation after selecting and crossing a single male offspring, to prevent genotyping females fertilised by a male transgenic for a different construct. Single flies were collected in PCR tubes containing 50 µl squishing buffer (10 mM Tris-HCL pH8, 1 mM EDTA, 25 mM NaCl, 200 µg/ml Proteinase K). Flies were disrupted in a Bead Ruptor (Biovendis) for 20 s at 30 Hz. Samples were then incubated for 30 min at 37°C, followed by heat inactivation for 3 min at 95°C. 3 µl of supernatant were used in 30 µl PCR reactions with primers pCFD6seqfwd2 and pCFD6seqrev2. PCR amplicons were analysed by Sanger sequencing with primer pCFD6seqrev2.

## Selection of lethal and viable target genes

Genes considered 'known lethal' or 'known viable' were chosen based on information available in FlyBase (release FB2018_1). For each gene report we manually reviewed the lethality information available in the phenotype category. We did not consider information based on RNAi experiments, as these typically were performed with tissue-restricted Gal4 drivers and residual expression might mask gene essentiality. Annotations of viability in FlyBase are heavily skewed towards lethal genes, likely reflecting the uncertainty in many cases whether a viable phenotype reflects residual gene activity of a particular allele.

## Immunohistochemistry

Immunohistochemistry of wing imaginal discs was performed using standard procedures. Briefly, larva were dissected in ice cold PBS and fixed in 4% Paraformaldehyde in phosphate buffered saline (PBS) containing 0.05% Triton-X100 for 25 min at room temperature. Larva were washed three times in PBS containing 0.3% Triton-X100 (PBT) and then blocked for 1 hr at room temperature in PBT containing 1% heat-inactivated normal goat serum. Subsequently, larva were incubated with first antibody (mouse anti-Cas9 (Cell Signaling) 1:800; mouse anti-Cut (DSHB, Gary Rubin) 1:30; guinea pig anti-Sens (Boutros lab, unpublished) 1:300; rabbit anti-Evi [*Port et al., 2008*] 1:800) in PBT overnight at 4°C. The next day, samples were washed three times in PBT for 15 min and incubated for 2 hr at room temperature with secondary antibody (antibodies coupled to Alexa fluorophores, Invitrogen) diluted 1:600 in PBT containing Hoechst dye. Samples were washed three times 15 min in PBT and mounted in Vectashield (Vectorlabs). To visualize apoptotic cells wing discs expressing the apoptosis sensor GC3Ai (*Schott et al., 2017*) was fixed in 4% PFA, washed in PBT containing Hoechst and mounted in Vectashield.

## Image acquisition, processing and analysis

Images were acquired with a Zeiss LSM800, Leica SP5 or SP8 or a Nikon A1R confocal microscope in the sequential scanning mode. Samples that were used for comparison of antibody staining intensity were recorded in a single imaging session. Image processing and analysis was performed with FIJI (*Schindelin et al., 2012*). For the comparative analysis of anti-Cas9, GC3Ai and anti-Evi fluorescent intensities presented in *Figure 2* raw image files were used to select the wing pouch area and measure the average fluorescence intensity. Experiments were performed at least twice and more than three samples were analyzed for each experiment.

To produce the overlay of several wing imaginal discs shown in *Figure 1* the Fiji plug-in bUnwarpJ (*Sorzano et al., 2005*) was used. Images were rotated and cropped such that wing discs were oriented dorsal up and anterior left and positioned in the center of the image. A representative image was selected as 'target' and all other images registered to this target using bUnwarpJ, selecting 'mono' as registration mode and setting landmark weight to 1. Landmarks were manually selected around the outline of the target wing disc, as well as along the folds in the hinge region of

the disc. Registered images were then transformed to a binary image using the Fiji threshold function and assembled to an image stack. Shown are average intensity projections of the indicated number of images using the Fire lookup table. In the resulting image bright areas are CIGAR positive in many discs, while dark areas are devoid of CIGAR signal in most discs.

## Sequence analysis of CRISPR-Cas9 induced mutations

To determine the mutational status at each sgRNA target site the locus was PCR amplified and PCR amplicons were subjected to sequencing. To extract genomic DNA, flies were treated as described above under 'Genotyping of sgRNA flies'. Primers to amplify the target locus were designed to hybridize 250–300 bp 5' or 3' to the sgRNA target site and are listed in *Supplementary file 4*. PCR products were purified using the PCR purification Kit (Qiagen) according to the instructions by the manufacturer and sent for Sanger sequencing. While Sanger sequencing is less accurate and quantitative than deep sequencing of amplicons on, for example, the Illumina platform, it typically allows to cover both sgRNA targets on a single amplicon, which is necessary to account for mutations that result in deletions of the intervening sequence. In cases where this was not possible, for example due to the presence of a large intron between the target sites, each site was analysed on a separate PCR amplicon. To account for deletions in these cases additional PCR reactions containing the distal fwd and rev primers were included. Sequencing chromatograms were visually inspected for sequencing quality and presence of the sgRNA target site and analysed by Inference of CRISPR Edits (ICE) analysis (*Hsiau et al., 2019*).

## Acknowledgements

We would like to thank David Ish-Horvitz, Tony Southall and Norbert Perrimon for discussions and Maja Starostecka for helpful comments on the manuscript. We would like to acknowledge Erich Brunner and Konrad Basler for sharing material prior to publication. We would like to thank Lelia Wagner, Ainoa Tejedera and Christina Schlagheck for technical assistance and Sandra Müller (Teleman lab, DKFZ) for microinjections. We are also grateful to Kadri Oras and Simon Collier (Fly Facility, University of Cambridge) and Deepti Trivedi Vyas *Drosophila* Facility, NCBS, Bangalore) for *Drosophila* transgenesis. We were supported by the DKFZ Light Microscopy Core Facility, the Zeiss Application Center at the DKFZ and the Nikon Imaging Center at Heidelberg University. Work in the lab of MB is in part supported by grants from the European Research Council (ERC) and the DFG (SFB/TRR186, SFB1324).

## Additional information

### Funding

| Funder | Grant reference number | Author |
| --- | --- | --- |
| Deutsche Forschungsgemeinschaft | SFB/TRR186 | Bojana Pavlović<br>Michael Boutros |
| Deutsche Forschungsgemeinschaft | SFB1324 | Michael Boutros |
| European Research Council | DECODE | Michael Boutros |

The funders had no role in study design, data collection and interpretation, or the decision to submit the work for publication.

### Author contributions

Fillip Port, Conceptualization, Data curation, Formal analysis, Supervision, Investigation, Visualization, Writing - original draft, Project administration, Writing - review and editing; Claudia Strein, Mona Stricker, Celine Beyersdörffer, Amy Hess, Katharina Kern, Laura Lange, Nora Langner, Roberta Malamud, Lukas Schmitt, Lukas Voos, Investigation, generated the sgRNA library; Benedikt Rauscher, Florian Heigwer, Software, Investigation, Methodology; Jun Zhou, Investigation, Methodology; Jana Frei, Bojana Pavlović, Kristin Rädecke, Investigation; Erica Valentini, Software, Investigation, generated essential IT infrastructure; Michael Boutros, Conceptualization, Formal analysis,

Supervision, Funding acquisition, Investigation, Writing - original draft, Project administration, Writing - review and editing

## Author ORCIDs
Fillip Port (iD) https://orcid.org/0000-0002-5157-4835
Florian Heigwer (iD) http://orcid.org/0000-0002-8230-1485
Jun Zhou (iD) http://orcid.org/0000-0002-2101-9304
Michael Boutros (iD) https://orcid.org/0000-0002-9458-817X

## Decision letter and Author response
Decision letter https://doi.org/10.7554/eLife.53865.sa1
Author response https://doi.org/10.7554/eLife.53865.sa2

## Additional files

### Supplementary files
- Supplementary file 1. HD_CFD Drosophila lines 2019.
- Supplementary file 2. Results of the lethality screen.
- Supplementary file 3. Uncharacterized essential genes.
- Supplementary file 4. Oligonucleotide sequences.
- Supplementary file 5. Drosophila strains used.
- Transparent reporting form

### Data availability
All materials are available upon request. All data are contained within the manuscript and associated supplementary material. Transgenic fly lines are available through the Vienna *Drosophila* Resource Center (https://stockcenter.vdrc.at). Plasmids will be made available through Addgene (Addgene plasmids 127382–127388).

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
