## [Decision Letter]

**Acceptance summary:**

A persistent challenge in unraveling endogenous gene function is having the tools to determine when and where genes, particularly essential genes, are functioning in vivo. In this manuscript, the authors leverage CRISPR-Cas9 gene editing reagents to create an impressive toolbox to interrogate gene function with tissue specificity in *Drosophila*. The authors' results are presented in an easy-to-read, straightforward manner, and the figures are visually appealing. Their use of uORFs of different lengths to regulation Cas9 expression is a nice application. While other similar tissue-specific CRISPR resources for flies have been recently reported (e.g. Meltzer et al., 2019 and Poe et al., 2019), the large size of the authors' transgenic gRNA library and its focus on targeting members of specific protein classes (kinases, phosphatases, transcription factors) will likely make the manuscript of interest to a broad audience.

**Decision letter after peer review:**

Congratulations, we are pleased to inform you that your article, "A large-scale resource for tissue-specific CRISPR mutagenesis in *Drosophila*", has been accepted for publication in *eLife*. The reviewers have some suggestions to improve the manuscript, but you can consider them at your discretion.

A persistent challenge in unraveling endogenous gene function is having the tools to determine when and where genes, particularly essential genes, are functioning in vivo. In this manuscript, the authors leverage CRISPR-Cas9 gene editing reagents to create an impressive toolbox to interrogate gene function with tissue specificity in *Drosophila*. The authors' results are presented in an easy-to-read, straightforward manner, and the figures are visually appealing. Their use of uORFs of different lengths to regulation Cas9 expression is a nice application. While other similar tissue-specific CRISPR resources for flies have been recently reported (e.g. Meltzer et al., 2019 and Poe et al., 2019), the large size of the authors' transgenic gRNA library and its focus on targeting members of specific protein classes (kinases, phosphatases, transcription factors) will likely make the manuscript of interest to a broad audience.

Reviewer #1:

The manuscript presented by Port et al. makes important contributions to the rapidly evolving CRISPR/Cas9 field and is in its current form – barring some minor issues – suitable for publication in *eLife*. In particular, the development of a "tunable" Cas9 gene variant is a welcome addition that elegantly solves the issue of Cas9 toxicity. I have no major issues that need to be addressed. All remaining issues relate to clarity and typos.

Reviewer #2:

General assessment: A persistent challenge in unraveling endogenous gene function is having the tools to determine when and where genes, particularly essential genes, are functioning in vivo. In this manuscript, the authors leverage CRISPR-Cas9 gene editing reagents to create an impressive toolbox to interrogate gene function with tissue specificity in *Drosophila*. The authors' results are presented in an easy-to-read, straightforward manner, and the figures are visually appealing. Their use of uORFs of different lengths to regulation Cas9 expression is a nice application. While other similar tissue-specific CRISPR resources for flies have been recently reported (e.g. Meltzer et al., 2019 and Poe et al., 2019), the large size of the authors' transgenic gRNA library and its focus on targeting members of specific protein classes (kinases, phosphatases, transcription factors) will likely make the manuscript of interest to a broad audience.

Major comments:

The utility of a gRNA library for tissue-specific knock-out ultimately relies on the efficiency and predictability of the gRNA strains to disrupt the expression of the target gene(s). The authors report that ~10% of the gRNA strains give rise to false-negative or false-positive results in a viability test. The authors do not go into too much depth to determine the reason for the unexpected effect on viability for each gRNA strain (which is fine for the scope of this study). However, given the likely application of the transgenic gRNAs for tissue-specific knock-out, it would be helpful to know whether such gRNAs would still be effective in eliminating gene function in a subset of tissues and/or later in development or adulthood. Could the authors test for a developmental or morphological phenotype with a "false-negative" gRNA strain(s)? For example, gcm (glia cells missing): is the gcm gRNA strain able to induce a loss of glial cells (or other phenotype) that is characteristic of gcm loss-of-function alleles? Also, given that the gcm gRNAs are scored as only moderately effective (Figure 4—figure supplement 1), have the authors determined whether there is any correlation between the ICE score and severity or frequency of a phenotypic effect? On a related note, have the authors determined whether a gRNA strain will produce equally robust phenotypes in different somatic tissues, such as the wing versus eye or nervous system or intestine (apologies if I missed this)? These questions are aimed at understanding how this collection of gRNA strains could be effectively used to systematically elucidate gene function in individual tissues.

---

## [Author Response]

A persistent challenge in unraveling endogenous gene function is having the tools to determine when and where genes, particularly essential genes, are functioning in vivo. In this manuscript, the authors leverage CRISPR-Cas9 gene editing reagents to create an impressive toolbox to interrogate gene function with tissue specificity in Drosophila. The authors' results are presented in an easy-to-read, straightforward manner, and the figures are visually appealing. Their use of uORFs of different lengths to regulation Cas9 expression is a nice application. While other similar tissue-specific CRISPR resources for flies have been recently reported (e.g. Meltzer et al., 2019 and Poe et al., 2019), the large size of the authors' transgenic gRNA library and its focus on targeting members of specific protein classes (kinases, phosphatases, transcription factors) will likely make the manuscript of interest to a broad audience.Reviewer #2:General assessment: A persistent challenge in unraveling endogenous gene function is having the tools to determine when and where genes, particularly essential genes, are functioning in vivo. In this manuscript, the authors leverage CRISPR-Cas9 gene editing reagents to create an impressive toolbox to interrogate gene function with tissue specificity in Drosophila. The authors' results are presented in an easy-to-read, straightforward manner, and the figures are visually appealing. Their use of uORFs of different lengths to regulation Cas9 expression is a nice application. While other similar tissue-specific CRISPR resources for flies have been recently reported (e.g. Meltzer et al., 2019 and Poe et al., 2019), the large size of the authors' transgenic gRNA library and its focus on targeting members of specific protein classes (kinases, phosphatases, transcription factors) will likely make the manuscript of interest to a broad audience.Major comments: The utility of a gRNA library for tissue-specific knock-out ultimately relies on the efficiency and predictability of the gRNA strains to disrupt the expression of the target gene(s). The authors report that ~10% of the gRNA strains give rise to false-negative or false-positive results in a viability test. The authors do not go into too much depth to determine the reason for the unexpected effect on viability for each gRNA strain (which is fine for the scope of this study). However, given the likely application of the transgenic gRNAs for tissue-specific knock-out, it would be helpful to know whether such gRNAs would still be effective in eliminating gene function in a subset of tissues and/or later in development or adulthood. Could the authors test for a developmental or morphological phenotype with a "false-negative" gRNA strain(s)? For example, gcm (glia cells missing): is the gcm gRNA strain able to induce a loss of glial cells (or other phenotype) that is characteristic of gcm loss-of-function alleles? Also, given that the gcm gRNAs are scored as only moderately effective (Figure 4—figure supplement 1), have the authors determined whether there is any correlation between the ICE score and severity or frequency of a phenotypic effect? On a related note, have the authors determined whether a gRNA strain will produce equally robust phenotypes in different somatic tissues, such as the wing versus eye or nervous system or intestine (apologies if I missed this)? These questions are aimed at understanding how this collection of gRNA strains could be effectively used to systematically elucidate gene function in individual tissues.

The reviewer is raising a very interesting point: To what degree are observations regarding the efficiency of conditional CRISPR mutagenesis from one condition transferable to another? For example, one might imagine that the mutation frequency at a given target site could vary between cell types or states depending on, for example, chromatin accessibility or differences in the expression of DNA repair enzymes. Systematic studies of such effects within an organism are currently lacking in any system and it will be important to perform such studies in the future. Another aspect is that the same mutation frequency will have different practical consequences in different assays. For example, a sgRNA lines with relatively low efficiency might cause bi-allelic gene disruption in only a small fraction of cells and this might not cause detectable phenotypes in one assay (e.g. viability), but easily detectable phenotypes in another (e.g. morphology of the appendages). We therefore believe that the false-positive and false-negative rate in conditional CRISPR experiments is likely going to be context dependent and are looking forward to more scientists adopting this technology and sharing their experience. Within this study we have tested conditional CRISPR mutagenesis in a range of different tissues (Figure 1) and have now added a paragraph in the Discussion highlighting work from other labs showing that is also works in postmitotic and endoreplicating cells.